# The 2019 Philippine UHC Act, Pandemic Management and Implementation Implications in a Post-COVID-19 World: A Content Analysis

**DOI:** 10.3390/ijerph19159567

**Published:** 2022-08-04

**Authors:** Maria Cristina G. Bautista, Paulyn Jean Acacio-Claro, Nori Benjamin Mendoza, Christian Pulmano, Maria Regina Justina Estuar, Manuel M. Dayrit, Vincent Edward Festin, Madeleine Valera, Quirino Sugon, Dennis Andrew Villamor

**Affiliations:** 1Graduate School of Business, Ateneo de Manila University, Makati 1200, Philippines; 2Department of Information Systems and Computer Science, School of Science and Engineering, Ateneo de Manila University, Quezon 1108, Philippines; 3Ateneo School of Medicine and Public Health, Ateneo de Manila University, Pasig 1604, Philippines; 4Department of Physics, School of Science and Engineering, Ateneo de Manila University, Quezon 1108, Philippines

**Keywords:** UHC implementation, content analysis, intraclass correlation, partnerships, pandemic response management, fiscal space

## Abstract

The 2019 Philippine Universal Health Care Act (Republic Act 11223) was set for implementation in January 2020 when disruptions brought on by the pandemic occurred. Will the provisions of the new UHC Act for an improved health system enable agile responses to forthcoming shocks, such as this COVID-19 pandemic? A content analysis of the 2019 Philippine UHC Act can identify neglected and leverage areas for systems’ improvement in a post-pandemic world. While content or document analysis is commonly undertaken as part of scoping or systematic reviews of a qualitative nature, quantitative analyses using a two-way mixed effects, consistency, multiple raters type of intraclass correlation coefficient (ICC) were applied to check for reliability and consistency of agreement among the study participants in the manual tagging of UHC components in the legislation. The intraclass correlation reflected the individuals’ consistency of agreement with significant reliability (0.939, *p* < 0.001). The assessment highlighted a centralized approach to implementation, which can set aside the crucial collaborations and partnerships demonstrated and developed during the pandemic. The financing for local governments was strengthened with a new ruling that could alter UHC integration tendencies. A smarter allocation of tax-based financing sources, along with strengthened information and communications systems, can confront issues of trust and accountability, amidst the varying capacities of agents and systems.

## 1. Introduction

The Philippines has taken great strides in moving towards universal health coverage. In 2019, the Philippine Universal Health Care Act (UHC), or Republic Act 11223, was signed and the planned implementation in January 2020 was disrupted by the COVID-19 pandemic [1]. This pause provides an opportunity to reflect on the provisions of the new law, in the light of the country’s pandemic response and overall health-system reforms. This study seeks to examine the UHC Act’s intentions and provisions, against the health-system structures tested during the pandemic. The goal is to contribute to a post-pandemic health system that is sufficiently agile to meet any new pandemic when it occurs. The 2019 Act was crafted to meet the universal health coverage (UHC) goals of effectiveness, quality, and affordability. The Act meets the goals and tenets of the global movement towards universal health coverage pursued by the World Health Organization (WHO) since 2010 [2]. Financial protection and equity are at the heart of the UHC Act. The overarching question that the study raises is whether its provisions is still ‘fit for purpose’, given health-sector performance since the pandemic began in 2020. At its core, the study seeks to examine for intentionality, ‘neglected’, and leverage areas that proved critical to the pandemic response management.

The content analysis approach is one widely used in political science to detect the policy positions of parties [3] and is often used in health policy systematic reviews [4,5]. The commonly known pillars of health systems analysis espoused by WHO, as well as the COVID-19 pandemic lessons’ frame, such as those identified by the British Academy in a multi-disciplinary evidence review [6], can be viewed as a priori considerations that serve as references for the legislation analysis. A scrutiny of a legislative product or law allows a forward-looking view on the reform intentions and/or the neglected and leverage areas critical in the law’s implementation. The authors are not aware of any similar work in legislative analysis in health policy.

This section includes an overview of the Philippine health system and the country’s pandemic performance, followed by the guiding framework.

### 1.1. Overview of the Philippine Health System

The Philippines is a middle-income economy in the southeast Asia region, with per capita gross domestic product (GDP) of 3550 USD in 2021 [7]. It has an estimated population of 111 million in 2021, with the median age of 25.7 years. Table 1 shows some salient features of the Philippine health-care system. Health expenditures comprised 5.6 percent of its gross national product in 2020, from 4.7 percent the previous year 2019 [8]. The Philippines national health accounts’ data showed that health expenditure by all sectors rose to 12.6 percent in 2019–2020, the first year of the pandemic, from 10.2 percent in the previous two years. Government schemes and compulsory contributory health care financing schemes comprised 45.7 percent of the source of spending, with household out-of-pocket payments at 44.7 percent, and the balance attributed to voluntary payment schemes. The year 2020 saw government and contributory financing schemes overtake household out-of-pocket payments, a welcome development for equity considerations [9].

Table 1 highlights the household preference for public medical facilities, which under the new UHC Act, must maintain 90% of beds as wards and 10% as private beds. The reverse is mandated for private facilities to maintain at least 10% as ward beds of its total bed capacity. The same Act requires no additional user charges be imposed on indigent and other special groups in the ward facilities. While health human resources were largely trained (86%) in fee-paying, private institutions, the new UHC Act expands public medical and health services training, with a return-to-service clause for the scholars of public universities and colleges. The public facilities had a larger share of nurses and midwives. In general, the medical doctors and nurses are found in hospitals, as opposed to primary care facilities.

Since the enactment, 60 operational guidelines and policies have been drawn up to implement UHC, and in accord with its Implementing Rules and Regulations (IRR). Fifty-five (55) are considered as finalized and issued. Thirty-three (33) implementation sites are considered to be advanced implementation sites (AIS) [14]. The UHC implementation process is presented by the DOH in various roadshows as being built around integration. Below are the ways in which the implementation is described:

*Structural integration*—toward the creation of the city-wide and province-wide health systems; this means bringing together the inter-local health zones (ILHZ) which group municipalities into districts. This structural integration means having various service delivery structures at province, municipal, and district levels work together more closely. Primary health care, the lowest level of operations, will operate as the system gatekeeper, channeling patients through diagnostic systems at secondary levels and specialist and inpatient care through to tertiary and the apex, as needed;

*Managerial integration*—towards the strengthening of the Provincial Health Board and the City Health Board; these boards are the policymaking instruments for UHC, as well as the units that approve resource allocation and execute oversight of the implementation; this will entail the bringing together of representatives from the various offices of the Province, Municipalities, DOH, and the private sector, both for-profit and non-profit. This integration will involve the strengthening of the Provincial Health Office as the technical arm of the province in overseeing the hospitals (provincial hospitals, district hospitals), as well as the public health facilities (rural health units, barangay health units). It is conceivable that the private sector facilities will become part of the scope of the Provincial Health Office;

*Financial integration*—this means bringing together the resources from national government (internal revenues, PhilHealth remittances, etc.), resources from the province, and resources from the municipality into one or two Special Health Funds (SHF) which will be managed by the province and highly urbanized cities, respectively. From this Special Health Fund, allocations and disbursements will be made.

This strategy for UHC implementation demonstrates a centralizing approach, as presently the provinces and highly urbanized cities are the owners and the locus of the health services delivery of the health system, responsible for planning, payroll, and budget allocations to government health services and activities in their jurisdiction.

When the 2019 UHC Act joint legislative deliberations began in 2017, relatively strong economic growth had been registered in prior years. Election promises and some reduction in the poverty levels have created a wider receptivity to health sector advocacies. The Act was discussed and drawn amidst much fiscal space, benefitting from payroll contributions and new revenue sources in the form of sin taxes (so called taxes on cigarettes, alcohol, and sugar content) and reforms that earmarked and increased the financial resources for health. The allocation of roles and responsibilities, where the Philippine Health Insurance Corporation (PhilHealth) is responsible for personal health services, while the Department of Health (DOH) covers for the population health services, is a convenient way to divide fiscal resources. The role of local government units (LGUs), who are the de jure owner of the facilities below regional levels, such as provincial hospitals, city and rural health centers, and barangay health stations, was not clearly specified. Many of the LGUs were not aware of their roles, as voiced in the deliberations about its implementation rules. A separate law, the Local Government Code (LGC) or Republic Act 7160 in 1991, governs LGUs. The integrated local health systems (ILHS), established within the provincial and highly urbanized cities, are tasked under the UHC Act to be responsible for both the planning and managerial supervision of population-based and individual-based health services within respective local jurisdictions (Sec. 19.8).

One area affecting national–local relationships is the health workforce. When local governments took over the health functions in the 1991 decentralization, they had to contend with paying health personnel with a higher pay structure, different from regular government personnel, and mandated in another law, the Magna Carta for Public Health Workers Act [15]. The differences in pay structure were justified in terms of hazards, subsistence, longevity pay, laundry, and remote assignments, along with performance-based allowances. The pay and benefit structures are not widely adopted in the private health sector. A comprehensive policy review culled 134 policies consisting of 73 laws and 61 executive policy issuances of different government agencies affecting human resources for health [16]. The payment for the health workforce is a contentious point for private providers, particularly, if this will be part of the requirements for accreditation. The private sector may not be able to pay its workers all of the benefit payments given to public health care workers. The health workers paid from the local government funds are not receiving the same amount as the national government-funded health workers, creating a contentious area within the same workspace.

In a recent policy development, known as the Mandanas Ruling by the Supreme Court in 2018 (and confirmed in 2019), the LGUs’ share of national funds or the Internal Revenue Allocation (IRA) will increase by an estimated 30.7% under the new funding formula, without earmarks. The increase was made possible as the allocations were expanded from a broader set of taxes, including customs and excise taxes. The intention is to support full devolution, assisting the LGUs with their service delivery functions, including health services [17]. This puts the LGUs in a position of strength with respect to the two other institutional custodians of UHC—the DOH and PhilHealth.

### 1.2. Pandemic Performance

By mid-February 2022, the 24th month since the pandemic was officially declared, the Philippines had registered 3.6 million cases and 55 thousand people had died from COVID-19. The active cases, tested via RT-PCR, at that point in time were 72.3 thousand. An estimated 62% of the population were officially reported by then to have received at least one dose of the vaccine [18]. There was no vaccination for children below 12 years old at this point in time. The Philippines ranked bottom of the Bloomberg COVID-19 resilience score in January 2022, due largely to the severity and duration of the lockdown periods, vaccination rates, openness to travel via flights, and travel routes [19].

The fiscal space for UHC implementation has since shrunk with the pandemic; the structure of service delivery and financing are challenged with the physical and human resource requirements and coordination needs. In exploring complex systems, such as health care, the country’s initial response during this pandemic can be characterized by dysfunctional coordination platforms, inadequate and mismanagement of resources, as well as limited information systems and capacities, as similarly indicated in systems thinking analysis [20,21]. That the country was able eventually to track cases, continue to monitor them, model, and anticipate trends was due to the coming on board of knowledge partners from academia and the private sector. Vaccine orders were made in advance by the private sector for their employees and eventually shared with the public sector, demonstrating regulatory partnerships.

The pandemic also highlighted the crucial and effective role performance of the LGUs, given weaknesses at the central level. Emergency funding was speedily approved nationally, but distribution to lower levels were not as quick. International partners (such as the Asian Development Bank) were also forthcoming with speedy support for infrastructure, such as laboratories and expensive GenXpert machines. The LGUs looked to the central authorities for guidance and support with materials. When these were unclear, not forthcoming, or delayed, they proceeded within their own mandates through their local legislative systems. Public–private partnerships were launched (Table 2). It is notable that the public counterparts were local governments (provincial governments and one highly urbanized city) and the University of the Philippines system. The imperatives of responding to the pandemic have seen additional resources poured into the health sector quite speedily, via legislation, private donations, and reallocation from other sectors. The absorptive capacities of some of the LGUs remain a concern [22]. It is interesting to note that these are hospital-based investments.

### 1.3. Framework

The WHO provides a framework where health systems’ performance is viewed in terms of three goals, four functions, and six building blocks [2]. Adapting from such a framework, this study’s view is one where healthcare is a complex system. It is composed of people and processes where the agents are defined by their roles, and corresponding scripts or actions (who does what) and processes facilitate relationships, so that each person or group is able to achieve specific goals (how it is executed) [24]. The way the roles and actions by the agents or actors in the system are undertaken is indicative of the type of relationships that enable certain activities to be pursued or not, and thus is determinative of the paths taken towards implementation and impact.

Figure 1 shows the key agencies, namely, DOH, PhilHealth, LGUs, and other stakeholders (private providers, insurers, community) at the center. Four health system building blocks relate to health service delivery (supply side), population or health promotion services, human resources, and stewardship. The functional component areas specific to UHC relate to financial protection, benefit package, payments and incentives, and contracts. Underpinning the reforms are the UHC values and goals of equity, efficiency, effectiveness, quality, and responsiveness. These are not shown, to avoid complicating the diagram. Ultimately, UHC intends to improve the health and well-being of Filipinos.

To assess the UHC Act from the lens of the experience under the pandemic, studies have identified interconnected areas that can be built on and are likely to endure for the long term: the strength of engaging with local communities; protection of health workers and other essential workers; effective national coordination; and transparent and consistent information or messaging [6,25]. Having resources and platforms to secure supplies and tools were necessary but not sufficient in the response. Examining whether the UHC Act has considerations in these areas are under discussion in this study.

## 2. Materials and Methods

A team composed of seven individuals/raters conducted content analysis of the UHC Act (RA 11223). These raters had legal, economics, physics, medical, public health, and computer science backgrounds. They were chosen for their varying knowledge of the UHC Act and general legislative processes to balance familiarity bias. One once served as PhilHealth Vice President, three read the Act in full for the first time, and two were fully acquainted with the Act but new to the given format. All of the team had graduate degrees. The instruction was to ‘tag’ clauses to the actors, blocks, and functions as read and not to go too deeply into ultimate ends or causes. The chapters and clauses of the law were put as rows in an Excel spreadsheet. The first step in the manual tagging of each chapter and clauses in the Act was made based on its directionality, or who it was being directed to. The next step was to examine these clauses into the functions and building blocks, using pre-determined components supporting the study’s framework (Figure 1). The taggers separately submitted their sheets before a compiled sheet was made. The areas tagged the most were examined for their implications. Items with the least tags were considered indicative of the areas requiring some attention for their significance to implementation.

This ‘tagging’ of the Act’s clauses and provisions to the framework components were further examined to check for reliability or consistency of the responses. The reliability or homogeneity of the measurement of the tagging process on different components performed by seven different raters was assessed through the intraclass correlation coefficient (ICC). The ICC on individual and average ratings was estimated. The average ICC can be used when the teams of raters are used to rate a target (called a component in this study). When the unit of analysis is an average rating, the interpretation of the ICC pertains to average ratings and not individual ratings.

Koo and Li [26] recommended selecting the correct form of ICC based on the model, type (single rater/measurement or the mean of k raters/measurements), and the definition of the relationship to be measured (consistency or absolute agreement). This study used a two-way, mixed-effects model because the raters of interest were not independently sampled. Thus, rater was considered a fixed effect, while targets/components were the random effects. Since each rating by the multiple raters were used in the analysis, the type of ICC referred to the “mean of k raters”. The relationship measured in this study was the consistency-of-agreement, because it represents correlation when the rater is fixed. Under consistency of agreement, the scores are considered consistent if the scores from any two raters differ by the same constant value for all of the targets. This implies that the raters give the same ranking to all of the targets. Researchers have pointed out that the consistency of agreement is useful when comparative judgments are made about the objects of measurement in the ICC [27].

The ICC estimates and their 95% confidence intervals (CI) were calculated using STATA/SE version 15.1, based on a mean-rating (k = 7), consistency of agreement, and two-way mixed-effects model.

Following a rule of thumb for the researchers to obtain at least 30 heterogeneous samples and to involve at least three raters when conducting a reliability study, it was suggested that ICC values less than 0.5 are indicative of poor reliability; values between 0.5 and 0.75 indicate moderate reliability; values between 0.75 and 0.9 indicate good reliability; and values greater than 0.90 indicate excellent reliability [27].

## 3. Results

Each rater tagged the different statements per chapter in the UHC Act according to the 14 components identified in the framework (Table 3). Each statement can have multiple tags. A total of 195 clauses/statements within 11 chapters, with 46 sections spread across them, were tagged.

Table 3 below shows the top three components that were most evident, based on the total tags by the raters. These components pertain to the DOH/National government, PhilHealth and contracts/enforcement. It is noted that a high standard deviation for the mean number of tags on contracts/enforcement implied that the frequency of tags by the respondents was dispersed. The number of tags for this ranged from 11 to 70. The least tagged component was Human Resource/Workforce Support Systems followed by the LGU/DILG (Department of the Interior and Local Government), and Benefits Demand Side (including health benefit plans).

Under a mixed-effects model, the individual ICC estimated was 0.69 (95% CI: 0.50–0.86), showing a correlation between the individual tags, while the average ICC was 0.94 (95% CI: 0.87–0.98), showing a correlation between the average tags made on the same component. The consistency of agreement of the seven raters for the individual tags on each component implied a moderate reliability, while there was excellent reliability for the average tags made per component. The estimated coefficients were statistically significant (F(13, 78) = 16.41, *p* < 0.001), indicating that both the individual and average tags were randomly dependent on the type of component and fixed on the corresponding rater making the measurements.

## 4. Discussion

The dominance of the DOH and PhilHealth is as expected, as the law can be viewed as their mandate to access increased revenue resources and implement wide-ranging health sector reforms to achieve or progress towards universal health coverage. Contracting, which surfaced as the third most tagged, refers to an instrumentality or mechanism that will be exercised by the primary agents to reach other agents. The least tagged, or what was viewed as less attended to in the law, included health human resources, LGUs, and the demand side of health services that involves health benefits. Three themes emerged as crucial in the UHC implementation, particularly from the post-COVID-19 lens. These are the national–local interactions, stakeholder engagement and contracting, and the capabilities and tools for coordination. This section discusses the challenges and directions in these areas.

### 4.1. National-Local Interactions

The UHC implementation appears to have been envisioned as a strategy of re-centralization, given the context of integration. At present, the provinces and highly urbanized cities are the owners and locus of health services delivery of the health system; responsible for planning, payroll, and budget allocations to government health services and activities in their jurisdiction. The content assessment, that showed LGUs as having the least attribution or responsibility, points to a critical neglect.

Under the implementing rules and regulations [28], the local health system refers to “all health offices, facilities and services, human resources, and other operations relating to health under the management of the LGUs to promote, restore or maintain health” (Sec. 19.2). It is envisioned, under Rule 19.6, that the integration of the province and city-wide health systems shall be undertaken “through a mechanism of cooperative undertakings” (pursuant to Section 33 of the Republic Act 7160; The Local Government Code of 1991) [29].

DOH and PhilHealth, outside of headquarters, conduct their primary responsibilities in a ‘cooperative’ arrangement. This arrangement has made for a mixed implementation of the devolution enshrined in the local government code [12]. The new UHC Act, however, has no compulsory power to enforce cooperation between the LGUs in their zones, even as tasks, such as managerial and financial integration and the provision of the needed resources and support mechanisms to make the integration possible and sustainable (Sec. 19.9), are indicated. This highlights the basic vulnerability of national–local relationships, that as political administration changes, so can the strategies and priorities change. The Mandanas ruling mentioned in Section 1.1 has given added leveraging power to LGUs with the increased funding share of the internal revenue allocation (that is, their share of the national taxes). These are not earmarked funds and it will require negotiations with the primary UHC implementors. In terms of accountabilities, only PhilHealth has a quasi-judicial authority on certain matters. The DOH has limited administrative remedies with respect to decision-making in the interlocal health decision making by their boards. The recent developments in public–private partnerships highlighted in Table 2 above, showed LGUs leading the partnerships in creating new facilities. The strengthening of the financial position of LGUs can be viewed as re-devolution. The integration discussions which strengthen the role of DOH in local matters will need re-calibration.

It has been indicated in the first part of this paper that one contentious area in national and local relationships has to with the responsibilities and payment for the health workforce. For the implementation of the UHC Act, more work needs to be completed in the harmonization of various laws and policies, as they affect both the government workforces and the private provision of care. The health workers paid from the local government funds are not receiving the same amount as the national government-funded health workers, creating a contentious area within the same workspace. The private providers may not be able to pay its workers all of the benefit payments given to public health care workers, and using this as a precondition for accreditation may put off private providers from UHC implementation. In the midst of the pandemic, there were mounting complaints from the private providers about unpaid accounts, an issue that has always been lodged against the institution [30], and for which the new UHC Act has not provided greater compulsion for PhilHealth, nor redress for the providers.

### 4.2. Other Stakeholders and Contracting

Our content analysis has shown that engagements with the private sector and community groups were weakly ascribed in the Act, and contracting figured prominently as the main instrument to relate with stakeholders. In the Act’s provisions on the private sector, its participation in the integrated local health system, through a contractual arrangement with the province-wide or city-wide health systems, is encouraged.

Since public (tax) and member funds are involved, services are not just purchased by central authorities but will be underpinned by contracts, or formal agreements among government agencies and affected stakeholders, including public and private facility operators, and the health workforce. Schuhmann and Bautista [30] observed that, while incentives are mentioned 18× in the draft IRR, there are more than 40 counts of contracts or contracting being mentioned in the implementation rules. With DOH and PhilHealth being hierarchical or bureaucratic organizations, contracting has been viewed largely as transactional, and the contracts are used to justify the transfers of resources. One of the biggest scandals during the pandemic was related to the government procurement of personal protective equipment (PPE), masks, and similar items [31].

The new UHC Act and its IRR view contracting as a “meeting of the minds”; however, among a very diverse group of parties with equally varying interests and considerations, this is not sufficient. It will be a challenge for the DOH, DILG, PhilHealth, and the LGUs to develop a contract that will satisfy the varying interests of the diverse group of entities sought to be brought together. The conceptual nuances of managing through contracts has been discussed in more depth [32]. Much of the language of the Act and the rules regarding contracts is directive, and for the individual-based health care provider networks, it appears that there is a requirement to form a single entity under a network. This is seen in the requirement of “networks exhibiting proof of legal personality” (Sec. 18.4.f). This clause forms a likely disincentive to private sector providers who are normally in competition with one another in small markets.

The DOH and PhilHealth have been given the task of providing general guidelines for these contractual arrangements, hence the impression and concern that these contracts will be very “top down” in nature is warranted. It is recommended that, to counteract this possibility, the task ahead is to empower and strengthen the Provincial and City Health Boards in order to provide nuanced, localized, and, consequently, responsive oversight and management over the contracts that establish the ILHS and other networks.

The role of non-government organizations, patient, and community groups was highlighted during the pandemic. They were involved in community mobilization, contact tracing, and coordinating the treatment of individuals in the community, even the provision of social assistance and support. They may continue to be partners post-pandemic. Access to UHC funds is likely also to be through the contracting mechanism. Government procurement systems are not user friendly to community groups and non-government organizations. Membership in the health boards is one venue for participation.

### 4.3. Other Cross-Cutting Considerations

The top-down approach has created various anomalies in the light of limited tools utilized to implement the new Act. Moving contracts from being mere transactions to moving monies and paying vendors will require better provider payment systems, which rests on improved information technology systems; with both, steering the system towards health care quality. Financing and information systems form the backbone on which the structure of the health system stands. The new UHC Act’s central feature divided responsibilities by type of intervention; between DOH for population health activities and PhilHealth for personal health care. Linkages on financing, particularly on payments to providers, did not prominently figure in the content assessment.

Fair provider payment processes are critical to push for quality services from accredited hospitals, using analytic and evidence-based approaches, rather than relying on its quasi-judicial mandate and flexing its muscle on fraud detection and conflicting policies. The funds from these two agencies will be pooled in provincial and highly urbanized cities to form the special health fund that is managed by reconstituted local health boards. Considering the slight majority of the private sector over government health in the facilities count, there is no indication as to how the private sector will accept not being directly in receipt of funds from the main insurer, PhilHealth. Instead, they will have to collect against the Special Health Fund (SHF), which is managed by the local political bureaus. Furthermore, the payment delays to hospitals, scandals over inappropriate payments, and manipulation of the system through case creep (how one simple case diagnosis will be upgraded to a more complicated case) have percolated into trust issues between PhilHealth and the private providers [33].

While the UHC law improves on the governance of PhilHealth by tightening the qualifications of PhilHealth board members, the financing reforms needed would require more transformative leadership. The pandemic experience has shown how easy it is to wipe out the financial reserves saved over the surplus years, erode agency credibility with the scandals and untimely premium increases. Cost efficiency is gained by altering the way payment is made to the providers, by moving from case payments to more cost control systems, such as global budgeting [34]. The Act’s Chapter IV, Section 18-B mentions performance-based, close-end, prospective payments based on diagnostic categories. This will require capacities, technical skills, and information that may not be currently available. Budget transfers, outside of premium revenues, by members will require better budget cycle planning and financial reporting.

The information requirements and systems needed for efficiency and equity are vast. While new regulatory agencies have been identified in the new UHC Act, such as the Health Promotion Bureau and Health Technology Assessment, the availability and quality of the data and information they will be working with for improved decision-making still need to be constructed. As they generate the data, frequent analysis and updates on diagnostic codes, weights, and costs will form part of the information technology systems that will yield more transparent and robust internal evaluation, utilization processes, knowledge exchange, and management for a nimbler health system.

As decentralized local delivery systems and a strong private health sector are the contexts under which the centralizing approach of the new UHC will be undertaken, incentives will be the main mechanism by which purchasing agencies, the DOH, and PhilHealth, are going to relate with stakeholders. Under a strengthened central authority, incentives need to evolve from simple ‘carrots’ or bargaining sticks to being part of a global budget scheme, similar to how international agencies, such as the Global Fund, allocate funds to countries. It is based on robust modeling and evaluation protocols. Only through an interface with technology and evidence-based management can the financial flows in the system be used to drive the strategic choices of users, be it as a provider, health center staff, patients, managers, or decision makers. Only then will the system be efficiently and equitably managed to ensure affordability and universal coverage. Without innovations, adoption will be slow, if not resisted.

A key challenge for the information system sector is the provision in the UHC Act’s implementing rules and regulations, stating that the Department of Health and the Philippine Health Insurance Corporation will be funding the development and upgrading of the information system software to be utilized by health care providers and insurers, at no cost (Section 36.3) to them. Several initiatives have been undertaken through the Philippines’ eHealth Agenda framework of 2010. Some of the key players in the pandemic case reporting systems were from the group. Central systems must be designed with local capacities and interoperability in mind. Knowledge partnerships have been shown to be critical during the pandemic, and can only be strengthened with less hierarchical systems.

## 5. Conclusions

This content assessment study set out to examine the provisions of the new UHC Act for its intentions and directionality. While the ICC estimates obtained in this study implied generally good reliability, these results may not be generalized to other populations of raters, as the raters were not independently sampled but rather fixed in this mixed-effects model. The educational and professional backgrounds of the raters may have influenced the consistency of their ratings. While representativeness to the general population may be unwieldy with the method chosen, the analyses are meant to draw insights into law making in the health sector and for reflection on implementation design.

Drawing from various viewpoints and levels of technical expertise, a robust content analysis uncovered tendencies towards centralization. Weak system links were identified with respect to the health human workforce, financing and information systems, pointing largely to basic governance issues. The key players in the system, such as local governments and the private health providers and community groups, which played critical roles during the pandemic, are cursorily treated. The harmonization of the critical laws affecting the health workforce and the local government code is in order. The enhancement of devolution through additional non-earmarked funding for local governments is expected to create further tensions at implementation, in light of the centralizing tendencies of the new UHC Act. Weak relationships with local government units, the private sector, and community group engagements will need reexamination, given their importance in pandemic resilience and in the networked systems envisioned for UHC. In particular, a re-examination of the integration approach, in terms of the technical capacities for planning and evidence-based management in the light of fresh resources, may be in order. Viewing the UHC Act from the perspective of the pandemic also highlighted leverage areas to include innovations in governance areas, such as public–private partnerships, improved financing through provider payments’ reforms, and the smarter allocation of tax-based sources. Contracting and allocation arrangements can improve the quality and efficiency of decision-making in the UHC system. An alternative view of contracts and incentives beyond a mere transaction exchange towards system steering can be examined more empirically and contribute to a more equitable system.

## Figures and Tables

**Figure 1 ijerph-19-09567-f001:**
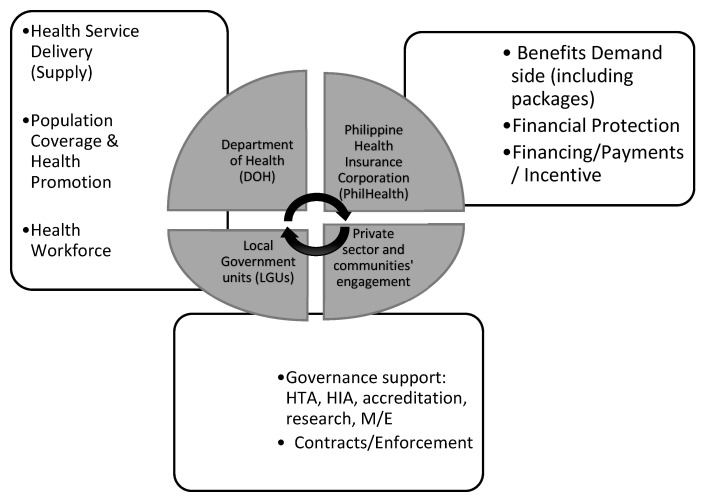
Framework of the Content Study on the Philippine UHC Act.

**Table 1 ijerph-19-09567-t001:** Philippine Health System: Overview of Basic Statistics.

Features	Figures (as Indicated)		
Current Health Expenditures-CHE (NHA 2020 in [8])	5.6% of GDP	Growth Rate of (CHE)	
		2019–2020	12.6%
		2018–2019	10.2%
Health financing (NHA: 2020 in [8])-households (out of pocket)	44.7%	Government and Compulsory Schemes	45.7%
Distribution of Household Out of pocket Expenses (NHA: 2020 in [8])	43.8% hospitals	28.2% drugs, pharmacies	9.0% preventive care
UHC Coverage Index (WHO and World Bank data in [10])	55%		
HC Utilization by households (DHS: 2017 in [11])	8% of household population sought care past 30 days	59% sought from public medical facilities	40% sought care from private medical facilities
Health service delivery (Dayrit et al., 2018 in [12])	1224 Hospital facilities, 64% private and 35% public in 2016; 66% located in main island	2587 city/rural health centers, 20,216 village health stations In 2016	Two-thirds are level 1 hospitals, with 41 beds on average; 10% are level 3, with 318 beds on average
Health Human Resources (Philippine Statistical Yearbook/PSY, 2018 in [13])	83% of health and medical graduates in 2015–2016 were from private schools (HIT, 2018)	3131 doctors in govt service; 1875 Dentists; 5975 Nurses	17,112 Midwives
	Public sector employs 61% of nurses and 90% of midwives (HIT, 2018)		91% of Medical doctors and 74% of nurses work in hospitals (HiT,2018)
Health Governance	DOH as the overall policy setting of population-based care, and provision of regional hospital and specialist services; PhilHealth, the social insurance arm, as purchaser of personal-based care	LGUs as facility owners, managers and implementers of health programs and services	Local Health Boards as advisers to chief executives and local legislatures, with DoH representative

Abbreviations: DHS—Demographic Health Survey; DoH—Department of Health; NHA—National Health Accounts; CHE—Current Health Expenditures; GDP—Gross Domestic Product.

**Table 2 ijerph-19-09567-t002:** List of Pipeline Public–Private Health Infrastructure Projects (as of February 2022).

	Implementing Agency	Investment (Php)	J-Yen	USD	Status
Philippine General Hospital-Diliman	University of the Philippines	21.3 billion	47.9 trillion	414.92 million	before ICC for approval
Baguio General Hospital and Medical Center, Renal Center	LGU and DOH	470 million	1 billion	8.0 million	
Cagayan Valley Medical Center Hemodialysis Center	LGU and DOH	330 million	742 million	6.4 million	
Philippine General Hospital—PGH Manila Cancer Center	PGH	4.6 billion	10.3 billion	89.6 million	before ICC for approval
Mariveles Mental Health and Wellness Center	LGU and DOH				
Makati Life Medical Center	LGU-Prv	5 billion	11 billion	97.2 million	

LGU investments are in various stages of implementation. ICC refers to the Intergovernmental Coordinating Council chaired by the President as final approval body for big ticket infrastructure investments. Source: Adapted and updated from [23].

**Table 3 ijerph-19-09567-t003:** Average number of tags for 14 components in the UHC Act by seven respondents.

Component	Mean ± Std. Deviation	Minimum–Maximum Frequency
DOH/national government	55.9 ± 8.5	42–68
PhilHealth	50.9 ± 9.3	39–67
Contracts/Enforcement	46.5 ± 23.2	11–70
Governance Support (HTA, HIA, accreditation, research, ME) *	42.7 ± 11.6	21–60
Financing/Payments/Incentives	39 ± 6.5	26–45
Community/Engagement	34.7 ± 12.5	16–53
Supply: Service Delivery	29.9 ± 8.8	22–48
Private Providers/Other Partners, (e.g., HMOs, Fund Managers) *	25.0 ± 6.9	19–37
Population Coverage and Health Promotion	23.9 ± 4.5	17–29
Financial Protection	23.6 ± 10.8	9–38
Values/Principles/Ethics	18.4 ± 6.4	11–29
Benefits Demand Side (Including packages)	16.6 ± 9.9	3–28
LGU/DILG	13.4 ± 4.0	8–19
Human Resource (HR)/Workforce Support Systems	4.4 ± 2.9	1–8

* Abbreviations: HTA = health technology assessment; HIA = health impact assessment; ME = monitoring and evaluation; HMO = health maintenance organization.

## Data Availability

The data presented in this study are available on request from the corresponding authors.

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
