# Peer review of "The 2019 Philippine UHC Act, Pandemic Management and Implementation Implications in a Post-COVID-19 World: A Content Analysis"

_ijerph, 2022, doi:10.3390/ijerph19159567_

Round 1

Reviewer 1 Report

Comments on "The 2019 Philippine UHC Act, pandemic management and implementation implications in a post-Covid19 world: A content analysis (ijerph-1777410)"

A content analysis of the 2019 Philippine UHC Act was conducted to answer the research question: will the provisions of the new UHC Act enable an agile response to forthcoming shocks?  The intraclass correlation coefficient (ICC) and analysis of variance (ANOVA) models were applied to check for reliability and consistency of agreement among raters.

Main Comments

My main comments are related to the methodology used in the manuscript (Sections 2 and 3.)

[1] I would strongly recommend giving more details on the methodology used (Section 2).  There are 10 different versions of ICC.  Which one was used and why?  A brief summary of choosing "Model", "Type" and "Definition" (see, Koo and Li, 2016) will be helpful for the reader.

[2] (Page 7, Line 250: CA-ICC) The term "consistency-of-agreement ICC" may be confusing.  Judging from the degrees of freedom in the F statistic (Page 9, Line 286), the authors' chosen ICC is to measure "consistency", not "absolute agreements."

[3] The components in Table 3 were derived from the framework (Figure 1).  However, some of the components cannot be found in Figure 1, for example, "Community/Engagement."  How does "Values/Principles/Ethics." fit into the framework?

[4] The authors may stress the fact that a mixed-effects model was used and the results cannot be generalized to other raters.

Minor Comments

[1] (Page 2, Line 67): "gross national product" = GNP, not GDP.

[2] (Page 3, Line 84): "... 90% private and ... 10% private ..."  Should one of the two "private" be "ward"?

Reference

Koo TK, Li MY. A Guideline of Selecting and Reporting Intraclass Correlation Coefficients for Reliability Research. Journal of Chiropractic Medicine. 2016; 15(2): 155-63.

Reviewer 2 Report

This article provides insights into the 2019 Philippine UHC (Universal Health Care) Act, “also referred to as Kalusugan Pangkalahatan (KP), that is the “provision to every Filipino of the highest possible quality of health care that is accessible, efficient, equitably distributed, adequately funded, fairly financed, and appropriately used by an informed and empowered public,” to identify potential underrated and/or neglected areas when governments try to reply efficiently to pandemic response management’s issues after a worldwide health disruption such as that of COVID-19. See abstract, lines 18-33. After averaging annual growth of 6.4% from 20102019, the Philippine economy has been in fact hit hard by the coronavirus disease (COVID-19) pandemic and contracted by 9.6% in 2020 (see: https://www.adb.org/sites/default/files/project-documents/55105/55105-001-rrp-en.pdf, pag. 9). The UHC Act had therefore an overall target related to the reform of the functional efficiency and financial management in the Philippines’ health sector (see: https://thinkwell.global/wp-content/uploads/2020/05/PH-UHC-Law-Series_Brief-1.pdf), and it seeks to revitalize health care through a whole-of-system, whole-of-government, whole-of-society, people-centered approach. The reform is rooted in the HSRA (Health Sector Reform Agenda) launched in 1999 and its implementation framework, the FOURmula One (F1) for Health in 2005, other than in the rural health units’ (RHUs) alignment for accreditation by PhilHealth as of October 2010, and finally, following the requests of the worldwide MDGs (Millennium Development Goals), in the Aquino Health Agenda (AHA) as of 2013 (see: https://doh.gov.ph/sites/default/files/basic-page/aquino-health-agenda-universal-health-care.pdf; see also: https://doh.gov.ph/sites/default/files/health_magazine/DOHissue1.pdf).   

Section 1. Introduction introduces the Philippine UHC Act and an overview of the Philippine Health System (see also Table 1 at line 78). Important to be underlined, the new UHC Act expands public medical and health services training, with a return-to-service clause for the scholars of public universities and colleges. See especially lines 87-90. Currently in the Philippines, the provinces and highly urbanized cities are the owners and locus of health services delivery of the health system, responsible for planning, payroll and budget allocations to government health services and activities in their jurisdiction. The new UHC Act then targets to reverse this trend instituting a centralizing approach to health services. Local Government Units (LGUs)—the units managing health services in the Philippines— had suffered from weak health system information management and performance accountability until when the new UHC Act has integrated their duties with other important  units in the Philippine’s health sector. Philippine Health Insurance Corporation (PhilHealth) and the Department of Health (DOH) have been also enforced by the new UHC Act. COVID-19 emergency has highlighted how the Philippines structure of service delivery and financing were challenged with new physical and human resource requirements as well as coordination needs. See especially lines 171-181. A public-private partnership (PPP) was necessary in the Country to empower the LGUs and in general the health sector’s investments’ capabilities.

A list of Philippines PPP projects is shown at Table 2, line 196.

Section 2. Materials and Methods illustrates the ICC intra-class correlation the authors wish to obtain through a content analysis of the UHC. See especially lines 243-264. The authors have therefore collected 30 heterogenous samples and involved at least 3 raters have tagged the different statements per chapter in the UHC Act according to the 14 components are shown in Table 3, line 278, by authors. ICC have been obtained also by average ratings and not only by individual ratings. Individual tags’ procedure is detailed at lines 228-242. 

Section 3. Results illustrates the top three components most evident, based on total tags by raters (see also again Table 3, line 278). The authors’ outcome highlight three themes are crucial in the UHC implementation: i. national-local interactions, ii. stakeholder engagement and contracting, and iii. capabilities and tools for coordination. 

Section 4. Discussion is a comment to the UHC implementation. Philippines health authorities, such as PhilHealth, have to work hard to allow for a harmonization of public-private law and policies, as well as for benefiting workers from the reform. Content analysis done by authors on the themes of UHC Act has been useful then to encourage better contractual arrangements between private sector, central authorities, and health workforce. See especially lines 352-361. Some obscure law requirement in the health sector is highlighted to underpin how government procurement systems are not user friendly to community groups and non-government organizations up to date. In fact, better payment processes, as well as fraud detection’s provisions may efficiently enhance best practices and ameliorate in general the health agency credibility to the eyes of customers.     

Challenges faced by the Philippines authorities are shown at lines 441-450, especially interoperability and flexibility in the perspective integrated post-UHC Act’s health system.  

Key assessment on the integration approach is then highlighted by authors at Section 5. Conclusions, lines 452-473. 

CHANGE REQUEST: 

1.         Please solve the acronym of UHC also in the title of paper. Acronyms must be solved first time they appear in titles, main body, tables, figures, appendices, notes, and so on,

2.         Please provide a better Table 2. Yours seems the shot of Table from the Asian Development Bank. Re-do you yourself the table should you wish to keep it in your main body,

3.         The Figure 1 is blurred and partially legible. Please provide a better figure or delete it from text,

4.         Please add a bullet point and/or a numbered list to highlight the three themes at lines 298-300 are crucial in the UHC implementation: i. national-local interactions, ii. stakeholder engagement and contracting, and iii. capabilities and tools for coordination,

5.         Please justify the alignment of paragraphs when missing, such as lines 290-450.  

MAIN REQUEST:

1.         I do not see along your paper any clear explanation about the targets of UHC Act. Is it a regulation about the price setting for services, or a reform about patient’s selection and price-escalation? Through the raters have tagged the different statements per chapter in the UHC Act according to your 14 components are shown in Table 3, line 278, do you have targeted a kind of HTA (Health Technology Assessment) or better a HIA (Health Impact Assessment)?

Since you seem more interested to the law features of the new Philippines UHC Act, I would better rephrase all the paper, focusing since its starts on the degree of competition among purchasers and providers and choice of payment and price negotiation methods.

Your concern about “responsibilities and payment for the health workforce” (see lines 334-5) seems to arise the need of direct contributors for the healthcare provisions. Instead, payment for the indigents will be covered by the national government. How do we define population-based and individual-based services? Are they divided in such a way that ensures accountability of assigned agencies? What are the sources of funds for health financing under the UHC Act?

Has this any impact on the service delivery especially in hard-to-reach GIDAs (Geographically Isolated and Disadvantaged Areas)?

I think to reply yourself to other interesting questions about UHC Act (see: https://doh.gov.ph/sites/default/files/health_magazine/Approved%20UHC%20FAQs.pdf) may improve the reading of your paper.

With Kind Regards,

 References:

Asilian-Mahabadi, H; Khosravi, Y; Narmin Hassanzadeh-Rangi, N;
Hajizadeh, E; Behzadan, AH (2020). Factors affecting unsafe behavior in construction projects: development and validation of a new questionnaire. International Journal of Occupational Safety and Ergonomics. 26:2, 219-226, doi: 10.1080/10803548.2017.1408243.

Gulliford MC; Adams G; Ukoumunne OC; Latinovic R; Chinn S; Campbell MJ (2005). Intraclass correlation coefficient and outcome prevalence are associated in clustered binary data. J Clin Epidemiol. Mar;58(3):246-51. doi: 10.1016/j.jclinepi.2004.08.012. PMID: 15718113.

Masood M; Reidpath DD (2016). Intraclass correlation and design effect in BMI, physical activity and diet: a cross-sectional study of 56 countries. BMJ Open. Jan 7;6(1):e008173. doi: 10.1136/bmjopen-2015-008173. PMID: 26743697; PMCID: PMC4716153.

Pleil JD; Wallace MAG; Stiegel MA; Funk WE (2018). Human biomarker interpretation: the importance of intra-class correlation coefficients (ICC) and their calculations based on mixed models, ANOVA, and variance estimates. Journal of Toxicology and Environmental health. Part B, Critical Reviews, Aug;21(3):161-180. doi: 10.1080/10937404.2018.1490128. Available from: https://europepmc.org/article/med/30067478.

Round 2

Reviewer 2 Report

Dear authors, I have appreciated the amendments and the clearer discussion of your focal points on the Philippine UHC Act and the provision for a more equitable healthcare in your Country.

I wish you the best for your career in the Sector.

With Kindest Regards,